# Ominoxanthone—The First Xanthone Linearly Fused to a γ-Lactone from *Cortinarius ominosus* Bidaud Basidiomata. CASE- and DFT-Based Structure Elucidation

**DOI:** 10.3390/molecules28041557

**Published:** 2023-02-06

**Authors:** Alice Trac, Célia Issaad, Mehdi A. Beniddir, Jean-Michel Bellanger, Jean-François Gallard, Alexei V. Buevich, Mikhail E. Elyashberg, Pierre Le Pogam

**Affiliations:** 1Équipe “Chimie des Substances Naturelles” BioCIS, CNRS, Université Paris-Saclay, 17 Avenue des Sciences, 91400 Orsay, France; 2CEFE, CNRS, Université Montpellier, EPHE, IRD, INSERM, 1919 Route de Mende, CEDEX 5, 34293 Montpellier, France; 3Institut de Chimie des Substances Naturelles, CNRS, ICSN UPR 2301, Université Paris-Saclay, 91198 Gif-sur-Yvette, France; 4Process and Analytical Chemistry, Merck & Co., Inc., 2015 Galloping Hill Road, Kenilworth, New Jersey, NJ 07033, USA; 5Advanced Chemistry Development Inc. (ACD/Labs), 8 King Street, Toronto, ON M5C 1B5, Canada

**Keywords:** CASE, DFT–NMR, xanthone, fungus

## Abstract

The UHPLC–HRMS analysis of *Cortinarius ominosus* basidiomata extract revealed that this mushroom accumulated elevated yields of an unreported specialized metabolite. The molecular formula of this unknown compound, C_17_H_10_O_8_, indicated that a challenging structure elucidation lay ahead, owing to its critically low H/C atom ratio. The structure of this new isolate, namely ominoxanthone (**1**), could not be solved from the interpretation of the usual set of 1D/2D NMR data that conveyed too limited information to afford a single, unambiguous structure. To remedy this, a Computer-Assisted Structure Elucidation (CASE) workflow was used to rank the different possible structure candidates consistent with our scarce spectroscopic data. DFT-based chemical shift calculations on a limited set of top-ranked structures further ascertained the determined structure for ominoxanthone. Although the determined scaffold of ominoxanthone is unprecedented as a natural product, a plausible biosynthetic scenario involving a precursor known from cortinariaceous sources and classical biogenetic reactions could be proposed.

## 1. Introduction

In the frame of our efforts dedicated to the description of new fungal metabolites [1], the UHPLC–HRMS analysis of the DCM/MeOH (1/1, *v*/*v*) extract of *Cortinarius ominosus* attracted our attention. It appeared that this mushroom accumulated considerable amounts of a compound detected at *m/z* 343.0450, for an elemental composition of C_17_H_10_O_8_. Quite remarkably, this molecular formula returned no hit in the *Dictionary of Natural Products* [2], prompting us to undertake its isolation for subsequent structure elucidation. Even though this precise molecular formula had not been formerly described, it was reminiscent of octaketide-derived anthraquinonic pigments repeatedly reported to occur from cortinariaceous sources [3]. Mushrooms pertaining to the genus *Cortinarius* and, more saliently yet, to the *Dermocybe* subgenus, are known to produce a wide variety of hydroxylated anthraquinones, preanthraquinones and pyranonaphthoquinones that are responsible for the yellow, orange or red colors of their fruiting bodies [4]. The sought-after compound, namely ominoxanthone (**1**), could be straightforwardly purified by preparative HPLC from the crude extract.

## 2. Results and Discussion

A first difficulty with **1** is that it was sparingly soluble in most common NMR solvents. This was rather unexpected, as the numerous polyketide pigments extracted from *Cortinarius* spp. had their NMR spectra recorded in usual solvents. Among the different investigated solvents, TFA-*d* represented the best option to facilitate the dissolution and acquisition of NMR data. Alternatively, DMSO-*d*_6_ also afforded ^1^H and ^13^C NMR data of convenient quality, but the ^1^H NMR signals were too broad to disclose a satisfying HMBC spectrum.

The ^1^H NMR spectrum of **1** revealed three aromatic protons ((δ_H_ 7.47 (1H, s), 7.53, and 8.00)) with the two latter being *meta*-arranged (1H both, d, *J* = 2.3 Hz), an oxygenated methylene group (δ_H_ 5.82 (2H, s)), and a methoxy group (δ_H_ 4.34 (3H, s)). The ^13^C NMR spectrum, in conjunction with the HSQC spectrum, revealed the presence of three carbonyl groups, three aromatic methines, an oxygenated methylene, a methoxy, and nine nonprotonated aromatic carbons (including four oxygenated carbons). Along with its molecular formula, these NMR landmarks hinted that **1** should correspond to a methoxylated octaketide, consistent with most polyketides known from *Cortinarius* species [3]. Biosynthetically speaking, canonical octaketide-derived anthraquinones are assembled from a linear precursor comprising an acetate starter and seven malonate units [5]. The regioselective fungal cyclization pattern of the linear polyketide precursor, the so-called F mode, is expected to afford two odd-numbered side chains that result in the canonical endocrocin framework (Figure 1A), even though the frequent decarboxylation at C-2 often leads to a final metabolite revealing only one odd-numbered side chain [6]. When embarking on the structure elucidation of **1**, we thus hypothesized that the oxygenated methylene group should be located at C-3, on the A-ring (Figure 1).

The relative disposition of substituents could be deduced from ROE experiments that placed the methoxy group between the two *meta*-disposed aromatic protons, as further confirmed based on the COSY crosspeak between these latter two. As this spin system did not involve the methyleneoxy signal, it was supposed that these signals were related to the C-ring (Figure 1A). These spectroscopic features were indeed reminiscent of the C-ring of many polyketide pigments that often incorporate this substitution pattern, occupying positions C-5, C-6, and C-7 (Figure 1A), but C-5 and C-7 could not be assigned a specific position [5,7,8]. Taking into account both molecular formula requirements and the polyketide origin of **1** guided us to infer that a -OH group should be located at C-8, even though this hypothesis found no spectroscopic support. Even though the occurrence of an oxygenated substituent at C-8 is consensual within cortinariaceous polyketides, a comparison of C-ring NMR signals to those of anthraquinones disclosing a similarly substituted C-ring revealed a very limited degree of NMR spectroscopic data similarity. In particular, the difference in the ^13^C NMR chemical shifts of the aromatic methines of this ring in **1** contrasted with former anthraquinones that disclose nearly equivalent resonances for these two positions [5,7]. Regarding the other spin system, tentatively located on the A-ring, ROESY and COSY crosspeaks could be observed between the oxygenated methylenic protons H_2_-3′ and the third, isolated aromatic proton H-4. The HMBC correlations from the oxymethylenic protons H_2_-3′ to the carbonyl-type carbon resonating at δ_C_ 176.3 (2-COO) to two aromatic quaternaries at δ_C_ 108.6 (C-2) and 158.8 (C-3) and to the aromatic methine at δ_C_ 104.0 (C-4) defined a benzenoid ring fused with a γ-lactone nucleus. These NMR features evoked cortinariaceous metabolites such as noraustrocorticin (Figure 1B) [5]. The good agreement between these substructure NMR data and those obtained from noraustrocorticin further supported this option, also suggesting that a −OH group should be located at C-1, in conjunction with the requirement for this cycle to be pentasubstituted and with molecular formula requirements. Again, this assumption is reasonable from a biosynthetic standpoint but lacked direct spectroscopic evidence. Even after two phenolic groups had been introduced without direct spectroscopic support, three oxygen atoms still had to be placed onto the structure of **1** to fulfill molecular formula requirements. This rules out the possibility for **1** to disclose an anthraquinonic scaffold. Interestingly, the cursory examination of the HMBC spectrum revealed a ^4^*J* HMBC correlation from both the aromatic protons resonating at δ_H_ 7.53 (H-5 or H-7) and at δ_H_ 7.47 (H-4) to a carbonyl-type carbon at δ_C_ 184.1 (C-9), determining that **1** comprised a benzophenone frame. The most straightforward strategy to complete the structure, in agreement with the obtained molecular formula, was to connect the two benzene rings through a lactone function tethering C-4a with C-10a. Such a connection would result in an oxepinedione-type B-ring, affording a first structure proposition, **1A** (Figure 1C). This tentative core proves satisfactory for some spectroscopic features that were inconsistent with an anthraquinone core. Accordingly, the HMBC correlation from the aromatic proton H-4 to an oxygenated sp² carbon resonating at δ_C_ 163.4 (C-4a) was in line with the introduction of an oxygen atom contiguous to it, as was further supported by the ^4^*J* HMBC correlation from the H_2_-3′ to the same carbon. Likewise, the long-range heteronuclear correlations observed from C-ring signals appeared consistent with structure **1A**. A first aromatic proton detected at δ_H_ 8.00 (H-5 in this hypothesis) revealed an HMBC correlation with C-6 and with a carbonyl-type carbon detected at δ_C_ 175.2, seemingly ascribable to the lactonic C-10, while the second aromatic proton at δ_H_ 7.53 (tentatively, H-7) displayed reasonable HMBC crosspeaks with C-6 and with an oxygenated *sp*² carbon resonating at δ_C_ 162.30 (compatible with C-8), along with the formerly reported ^4^*J* correlation to C-9. Even though these data were fully accommodated by structure **1A**, we could envisage an alternative structure, **1B**, compliant with all aforementioned spectroscopic features (Figure 1D). This structure comprised a xanthonic scaffold and a carboxylic acid function at C-8. Again, the HMBC correlations between the aromatic proton detected at δ_H_ 7.53 (corresponding here to H-5) and the oxygenated *sp*² carbon detected at δ_C_ 162.3 would then represent a correlation with C-10a. Likewise, the HMBC crosspeak between the second aromatic proton at δ_H_ 8.00 and the carbon at δ_C_ 175.2 could be explained by an H-5 to 8-COOH correlation.

A bibliographic survey confirmed the considerable difficulties in distinguishing these two scaffolds, as such pairs of compounds display identical carbon connectivities, and thus similar sets of HMBC correlations [9]. Interestingly, oxepinedione-containing metabolites were claimed to have been isolated from fungal sources on several occasions, mostly from micromycetes [10,11,12]. It turned out that many such metabolites had been erroneously elucidated at first and had later been revised into carboxylic acid-containing xanthones [9,13]. As an illustration, the structure revision of wentiquinone C is outlined below (Figure 2).

The minor differences in the NMR data of xanthones and seco-anthraquinones had been stressed at least twice throughout the literature, resulting in batch structure revisions [9,13]. These revisions rely on chemical derivatization strategies, as they are based on the cursory analysis of NMR landmarks of methylated derivatives. Accordingly, the appearance of methoxyester related resonances (δ_C_ < 55 ppm) after methylation presumably indicates the initial presence of a free carboxylic acid that could, therefore, not be involved in the cyclization process, thereby indicating a xanthone core. In simpler cases, a methoxy ester function natively existed in the natural product, but its misinterpretation as an aromatic methoxy group sometimes led to mistaken assignments of the structure as an oxepinedione derivative instead of the correct methoxyester-containing xanthone [9,13].

Interestingly, **1B** exhibited a structural difference to **1A** analogous to that existing between oxepinediones and their revised xanthone (Figure 2). A feature in favor of **1B** over **1A** was the excellent agreement between the NMR spectroscopic data of **1** with that of a carboxylic acid-xanthone (Figure 3) [9,14].

Overall, the cursory examination of the NMR spectroscopic data of **1** guided us to privilege structure **1B** for ominoxanthone, yet the information available from the key NMR experiments acquired from this isolate is rather limited due to its proton-deficient nature, rendering the structure elucidation tortuous. The so-called Crews’ rule foresees that NMR-based structure elucidation is at high risk of being inaccurate when the ratio of H/C atoms is less than one, as it is the case here [15]. The scarce information available from the NMR data of such structures is likely to result in several possible candidates, and it is therefore important to ensure that all possibilities have been duly considered [16].

To verify our structural hypothesis, we elected to submit the NMR spectroscopic data of **1** to a Computer-Assisted Structure Elucidation (CASE) workflow [17], through the ACD/Structure Elucidator (ACD/SE) expert system [18]. Briefly, this program takes the benefit of the HRMS-deduced molecular formula and of the 1D (^1^H and ^13^C) and 2D (COSY, HSQC, and HMBC) NMR spectra to create Molecular Connectivity Diagrams (MCD). This graphically depicts all atoms and XH_n_ moieties, along with their chemical shifts and properties (valence, hybridization, and admissibility of a neighbor heteroatom). COSY-derived connectivities, diagnostic of a length of one C–C bond (^3^*J*_HH_), and HMBC correlations, presumably indicative of a distance of one or two C–C bonds (^2,3^*J*_CH_), are further embedded into MCD. The logic-combinatorial analysis of the data displayed in MCD allows for structure generation of all possible structural isomers for a given molecular formula and 1D and 2D NMR data. Spectral and structural-based filters are subsequently applied to retrieve a list of tentative candidates [17]. The choice of the most probable structures among the set of generated candidates is based on the prediction of their ^13^C NMR chemical shifts using three empirical methods: incremental, neural network, and HOSE code (Hierarchical Organization of Spherical Environments) [19,20], resulting in a usual accuracy of 1.6 to 1.8 ppm [17]. Following this step of chemical shift prediction, the different structures of the CASE output file are ranked in order of increasing discrepancy between experimental and predicted NMR chemical shifts, usually expressed as the average deviation. Higher accuracy quantum-mechanical calculations can be applied to some of the top-ranked CASE-derived structures if this workflow does not succeed in unambiguously distinguishing the correct structure [21,22].

The spectroscopic data for **1** acquired in TFA-*d* were used by the ACD/SE program to produce an MCD (Figure 4), from which the strict structure generation workflow could propose eight possible structures within 31 min (Appendix A).

These structures were ranked in increasing order of average deviation d_A_, d_N_ and d_I_ as shown in Appendix A. Three top-ranked structures supplied with automatically assigned chemical shifts are presented in Figure 5. The maximal ^13^C deviation is also provided for each structure. Gratifyingly, the first ranked structure was identical to structure **1B**. Quite interestingly, the alternative structure **1A** turned out to be the third preferred option of the CASE workflow (Figure 5).

Unfortunately, the average mean deviations obtained between experimental and predicted ^13^C NMR chemical shifts proved to be of the same order and were higher than typically observed differences in such analyses (viz. d_n_ < 3 ppm, with *n* = A, I and N). As ACD/Predictors [18] were trained on databases containing NMR spectra recorded in conventional solvents, it is rather probable that the discrepancies could be a consequence of the NMR data being acquired in TFA-*d*. That is a very rarely chosen NMR solvent, but known to exert a substantial influence on ^13^C NMR chemical shifts [23]. We wanted to assess if better results could have been obtained by comparing simulated spectra against NMR data acquired in DMSO-*d*_6_, even though the degree of information retrieved from these spectra was comparatively lower due to broader ^1^H signals. To compensate for the low level of correlations obtained in this solvent, we wished to adapt the MCD obtained from TFA-*d* to the signals observed in DMSO-*d*_6_. For this purpose, the assignments were ordered by decreasing chemical shift values in both NMR data sets and transposed accordingly, unless explicitly contradicted by spectroscopic features obtained in DMSO-*d*_6_. While this strategy does not guarantee the correctness of all assignments, we assumed that swapped signals upon solvent change should display close chemical shift values, so this strategy should result in moderate chemical shift errors. The MCD created from NMR data which were acquired in DMSO-*d*_6_ is shown in Appendix A. As a result of structure generation, 20 structures were obtained in 34 min. The three top-ranked structures are presented in Figure 6.

Again, **1B** turned out to be the preferred structure (Figure 6). Satisfactorily, the magnitude of deviations obtained from this more conventional NMR solvent lay in the typical range of those obtained with correct structures. It is worth underlining that the second preferred option, the angular xanthone, was the same as in the CASE workflow performed from the NMR data obtained using TFA-*d* as a solvent. We have welcomed this proposal with some skepticism, as such an angular xanthone did not respect the endocrocin framework constantly found in anthraquinone carboxylic acids reported from *Dermocybe*/*Cortinarius* mushrooms [3,24]. Calculation of empirical DP4 probabilities endorsed the priority of structure #1 (Figure 6): DP4_A_(^13^C) = 95.29%, DP4_N_(^13^C) = 99.5%, and DP4_I_(^13^C) = 100%. Nevertheless, additional confirmation of the best structure by DFT ^13^C chemical shift prediction is desirable even in this case [21,22]. The selection of the best structure obviously deserved a better confirmation, guiding us to undertake accurate DFT–NMR calculations to discriminate between the three best structures (Table 1 and Table 2). The summary of these QM calculations is provided in Appendix A. The DFT–NMR predicted ^13^C NMR chemical shifts unequivocally defined structure #1 (corresponding to the so-called structure **1B**) as the most probable structure, based on both minimal root-mean-square deviation (RMSD) and lowest maximal deviation (Max_dev) of ^13^C NMR chemical shifts.

The newly reported ominoxanthone represents the first example of a xanthone linearly fused to a γ-lactone nucleus. This isolate is reminiscent of anthraquinonic compounds that were previously reported from cortinariaceous sources [5]. Biosynthetic interconversions from anthraquinones to xanthones are of considerable generality [13,14,25], and as such, a demethylated analogue of noraustrocorticin can be supposed to provide a straightforward access to ominoxanthone (Figure 7). It is interesting to note that a xanthonic structure linearly fused to a pyranone ring, leprocybin, had already been reported from *Cortinarius cotoneus* by Steglich and co-workers [26]. Leprocybin also presented important structural similarities with an anthraquinone, dermolactone, that was reported a few years later by Gill and Gimenez [7].

## 3. Materials and Methods

### 3.1. General Experimental Procedures

Ultraviolet (UV) spectra were measured on a Lightwave II+ WPA 7126 V.1.6.1 spectrophotometer. Infrared (IR) spectra were recorded with a PerkinElmer type 257 spectrometer. ^1^H and ^13^C NMR data were recorded on a Bruker 700 MHz NMR spectrometer equipped with a 5 mm TXO cryoprobe using TFA-*d* or DMSO-*d*_6_ as solvent. The solvent signals were used as references. HRESIMS measurements used an Agilent 6546 Accurate-Mass Q-TOF hyphenated with a 1290 Agilent Infnity II LC system. The chromatographic system was fitted with a Zorbax RRHD Eclipse Plus C_18_ column (2.1 × 50 mm, 1.8 μm). Sunfire preparative C_18_ column (150 × 19 mm, i.d. 5 μm; Waters) was used for preparative HPLC separations using a Waters Delta Prep equipped with a binary pump (Waters 2525) and a UV-visible diode array detector (190–600 nm, Waters 2996). A silica 80 g Grace cartridge was used for flash chromatography with a EZPrep Combiflash chromatography flash apparatus. Chemicals and solvents were purchased from Sigma–Aldrich.

### 3.2. Fungal Material

Basidiomata identified in the field as *Cortinarius semisanguineus* were collected in the forest of “Rochefort-en-Yvelines” by Sidney and Jean-François Gallard and Pierre Le Pogam, Yvelines (78), France, in October 2021 (TPS agreement 995468). The first morphological identification (Pierre Le Pogam) was refined by ITS rDNA sequencing and phylogenetic analyses, which demonstrated that the collected material rather belongs to the sister species *C. ominosus sensu lato* (Jean-Michel Bellanger). Species boundaries within this *C. semisanguineus/ominosus* lineage are not easy to delineate, even with molecular tools, and the taxonomy of the complex is currently under revision (Bellanger et al., in prep). An herbarium voucher specimen (PLPA20211001) was deposited at Université Paris-Saclay.

### 3.3. Extraction and Isolation

The air-dried and powdered basidiomata (210 g) were extracted with a (1/1, *v*/*v*) mixture of DCM and MeOH (4 L, room temperature, twice). The crude extract (7.9 g) was filtered under vacuum and concentrated by rotatory evaporation. All this residue was fractionated by flash chromatography on a silica 80 g Grace cartridge using a gradient of DCM with increasing proportions of MeOH (1:0 to 0:1). The flow rate was set at 60 mL/min. The flash chromatography separation yielded 17 fractions (D1–D17) according to their TLC profiles, which were analyzed by HPLC–QTOF–HRESIMS. Fraction D14 (23 mg) was subjected to a preparative HPLC separation using a gradient of CH_3_CN−H_2_O with 0.1% FA (20:80 to 60:40 in 15 min) to afford ominoxanthone (**1**, 4 mg).

Ominoxanthone (**1**). Pale yellow amorphous solid. UV (MeOH) λ_max_ (log ε) 251.0 (3.40), 282.0 (3.18), 310.0 (sh, 3.00), 344 (2.90) nm; IR ν_max_ 3219, 2918, 1746, 1720, 1680, 1390, 1246 cm^−1^; ^1^H (700 MHz) and ^13^C (175 MHz) NMR in TFA-*d* and DMSO-*d*_6_, see Table 1; HRESIMS *m/z* [M+H]^+^ 343.0450 (calculated for C_17_H_10_O_8_, 343.04484). The MS/MS spectrum was deposited in the GNPS spectral library under the identifier CCMSLIB00010129052.

### 3.4. CASE Analysis

The CASE analysis was performed using the commercially available ACD/Structure Elucidator program v. 14.56 [18].

### 3.5. DFT Calculations

DFT calculations were carried out using the Gaussian 16 software package [27]. Molecular geometries of structures 1–3 were optimized with the inclusion of a DMSO molecule at the b3lyp/6-31+G(d,p) level of theory. Upon geometrical optimization convergence, a frequency calculation within the harmonic approximation was conducted at the same level of theory, and local minima were characterized by the lack of imaginary frequency. The chemical shifts were calculated at the mPW1PW91/6-311+G(2d,p) level with the inclusion of the polarizable continuum model for DMSO (scrf = (solvent = dimethylsulfoxide)). Chemical shifts were determined from the isotropic values using the following scaling factors for ^1^H: slope = −1.0580 and intercept = 31.7217; for ^13^C: slope = −1.0496 and intercept = 186.2534 [28].

## 4. Conclusions

We herein elucidated and validated, through CASE and DFT strategies, the structure of ominoxanthone as a new subtype of xanthone. The structure of this compound is strongly reminiscent of that of the anthraquinonic noraustrocorticin and austrocorticin, the first examples of anthraquinone being linearly fused with a γ-lactone moiety, which had also been isolated from a *Cortinarius* sp.

## Figures and Tables

**Figure 1 molecules-28-01557-f001:**
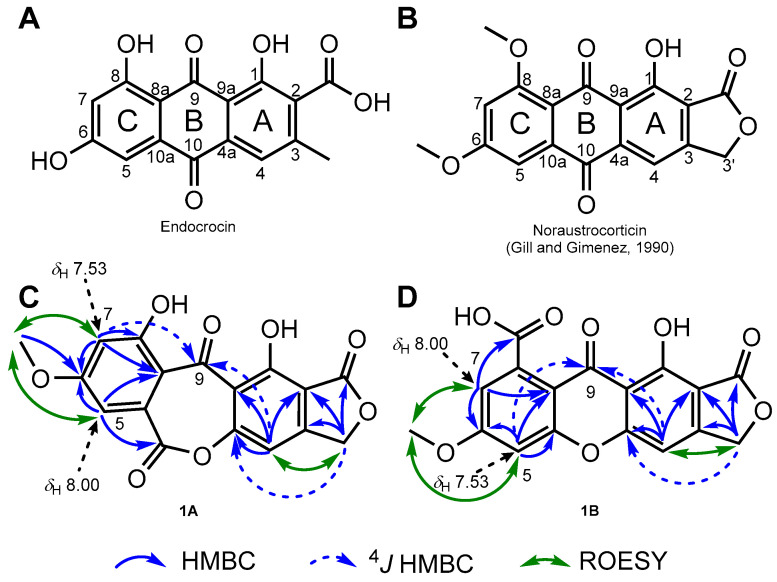
(**A**) Chemical structure of endocrocin, biosynthetic cornerstone of fungal polyketides. (**B**) Chemical structure of noraustrocorticin, anthraquinonic pigment reported from an unidentified Australian *Cortinarius* species. (**C**) Key 2D NMR correlations plotted on the oxepinedione-containing seco-anthraquinone scaffold hypothesis **1A**. (**D**) Key 2D NMR correlations plotted on the xanthone scaffold hypothesis **1B**.

**Figure 2 molecules-28-01557-f002:**
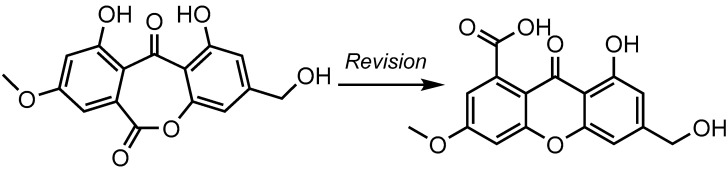
Structure revision of an oxepinedione-containing anthraquinone to a carboxylic acid-containing scaffold: example of wentiquinone C.

**Figure 3 molecules-28-01557-f003:**
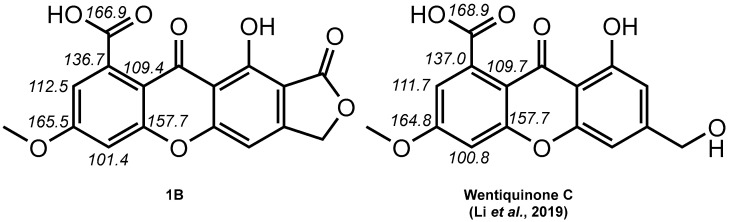
Compared ^13^C NMR chemical shifts related to C-ring signals of **1B** with those of wentiquinone C (the chemical shifts used for comparison are taken from the NMR spectra acquired in DMSO-*d*_6_).

**Figure 4 molecules-28-01557-f004:**
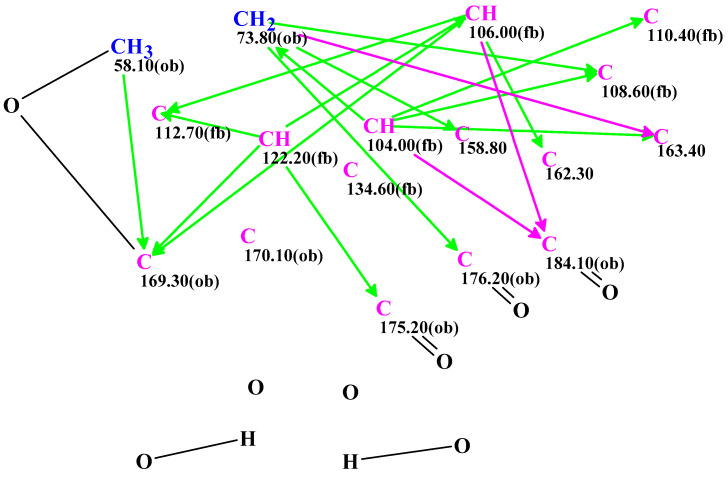
Molecular Connectivity Diagram obtained from the NMR data of ominoxanthone (C_17_H_10_O_8_) recorded in TFA-*d*. Carbons hybridized as *sp*^3^ and *sp*² are respectively written in navy blue and violet. The admissibility of a neighbor heteroatom is indicated by the labels « ob » (obligatory) and « fb » (forbidden). HMBC connectivities are indicated by green arrows. Violet arrows refer to nonstandard correlations (length *n* > 3, for ^n^*J*_HH,CH_). These correlations were set for the weak HMBC signals.

**Figure 5 molecules-28-01557-f005:**
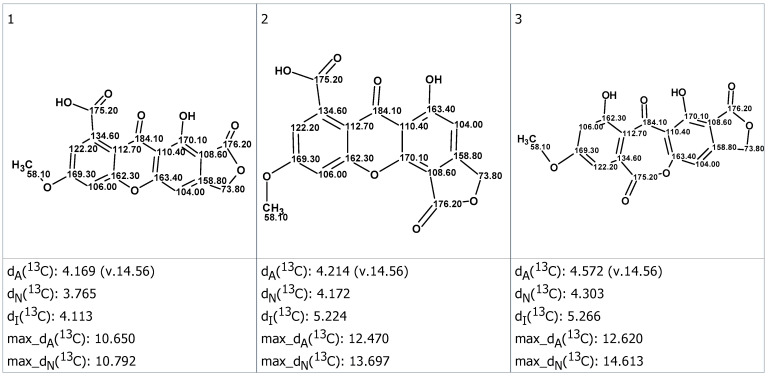
Three top-ranked candidate structures (1, 2 and 3) listed in order of increasing spectroscopic discrepancies. Average deviations are reported according to the prediction method—d_A_, HOSE-code; d_N_, neural network; and d_I_, incremental approach—along with the maximal ^13^C NMR chemical shift deviation for each prediction method.

**Figure 6 molecules-28-01557-f006:**
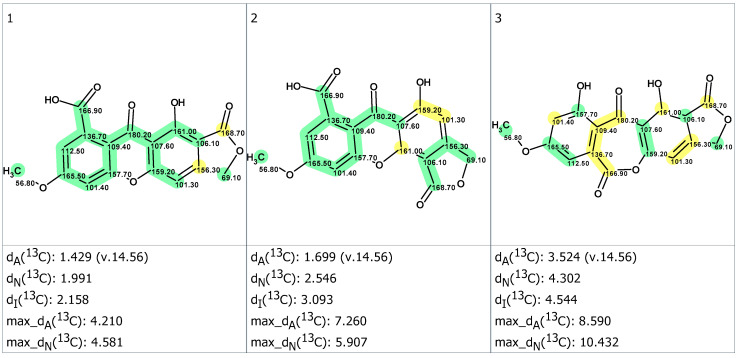
Three top-ranked candidate structures (1, 2 and 3) with ^13^C chemical shifts measured in DMSO-*d*_6_. Designations of deviations are the same as for Figure 5. The color code refers to the accuracy of the ^13^C NMR chemical shift prediction as follows: green = difference between theoretical and experimental < 3 ppm, yellow = difference comprised between 3 and 15 ppm, and red = difference over 15 ppm.

**Figure 7 molecules-28-01557-f007:**
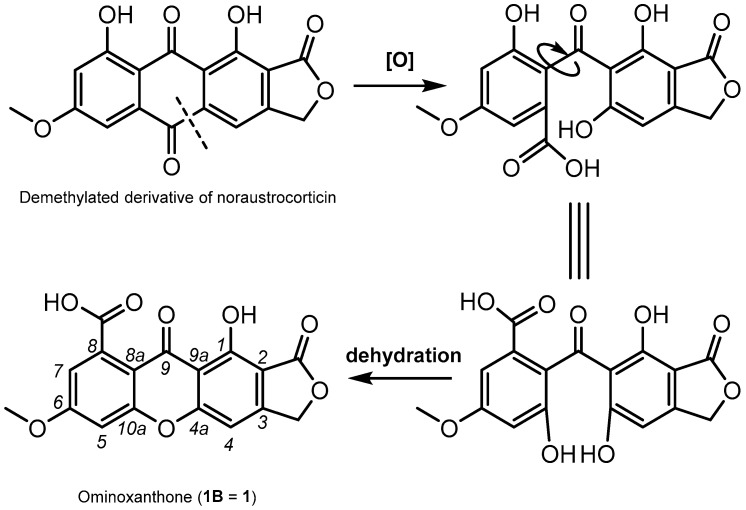
Plausible biosynthetic pathway for ominoxanthone (**1B**).

**Table 1 molecules-28-01557-t001:** ^1^H and ^13^C NMR (700/175 MHz) Spectroscopic Data for **1**.

	1 (TFA-*d*)		1 (DMSO-*d*_6_) *^a^*	
position	*δ*_H_, mult. (*J* in Hz)	*δ* _C_	*δ*_H_, mult. (*J* in Hz)	*δ* _C_
1		170.1		161.0
2		108.6		106.1
3		158.8		156.3
4	7.47 (1H, s)	104.0	7.25 (1H, br s)	101.3
4a		163.4		159.2
5	7.53 (1H, d, 2.2)	106.0	7.30 (1H, br s)	101.4
6		169.3		165.5
7	8.00 (1H, d, 2.2)	122.2	7.07 (1H, br s)	112.5
8		134.6		136.7
8a		112.7		109.4
9		184.1		180.2
9a		110.4		107.6
10a		162.3		157.7
2-COO		176.2		168.7
3-CH_2_O	5.82 (2H, s)	73.8	5.42 (2H, s)	69.1
6-OCH_3_	4.34 (3H, s)	58.1	3.98 (3H, s)	56.8
8-COOH		175.2		166.9

*^a^* Note that the NMR data acquired in this solvent are not secured by 2D NMR data, so some of the provided assignments may be inverted between.

**Table 2 molecules-28-01557-t002:** Experimental and DFT-calculated ^13^C NMR chemical shifts (ppm) for the three top-ranked candidate structures for ominoxanthone shown in Figure 5 and Figure 6 (for clarity, the provided numbering fits the best-ranked candidate; site-by-site assignments of the two other candidate structures are given in Appendix A).

Carbons	Exp. (DMSO)	1	2	3
1	161.0	161.46	151.87	161.23
2	106.1	105.27	102.44	108.32
3	156.3	156.89	157.33	157.22
4	101.3	99.43	103.36	103.88
4a	159.2	158.10	167.07	154.57
5	101.4	99.28	99.24	110.08
6	165.5	164.28	164.21	163.65
7	112.5	115.09	114.93	109.95
8	136.7	137.15	136.96	128.13
8a	109.4	109.32	109.57	114.74
9	180.2	177.70	176.91	191.89
9a	107.6	107.02	107.32	115.48
10a	157.7	156.47	155.91	163.10
2-COO	168.7	166.94	166.62	167.46
3-CH_2_O	69.1	68.39	68.68	68.48
6-OCH_3_	56.8	53.95	54.17	53.75
8-COOH	166.9	169.15	169.18	162.13
RMSD, ppm		1.66	3.59	5.48
Max_dev, ppm		2.83	9.12	11.64
R^2^		0.99921	0.99558	0.98980

## Data Availability

The data presented in this study are available on request from the corresponding author.

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
