# Peer review of "Ominoxanthone—The First Xanthone Linearly Fused to a γ-Lactone from Cortinarius ominosus Bidaud Basidiomata. CASE- and DFT-Based Structure Elucidation"

_molecules, 2023, doi:10.3390/molecules28041557_

Round 1

Reviewer 1 Report

The manuscript "Ominoxanthone the first xanthone linearly fused to a γ-lactone from Cortinarius ominosus Bidaud Basidiomata. CASE and DFT based structure elucidation" by A. Trac et al. is devoted to the isolation and structure elucidation of fungal metabolite. The described compound has very low H/C ratio and low-informative proton NMR spectra. The elucidation of the exact structure becomes a puzzle despite the low molecular weight. The work is written very clear and good illustrated. I guess the work is quite interesting, but it could not be recognized as a finished. The manuscript could be accepted only after serious revision. The main points of criticism are the follows:

1. So, may be it's my "chemist's bias" but I cannot agree with the authors main idea: NMR data + CASE and DFT calculations + some literature examples are enough for structure assignement. Without independent experimental proofs (crystallographic data or chemical derivatization results) it remains only "proposed" or "plausible" structure.

2. The section "Conclusions" should be added and the main points of the work should be summarized.

Minor corrections needed:

1. The format of the references inside the figures and schemes should be the same as used in the text. Please, change the Fig. 1B (Gill and Gimenez,1990), Fig. 3 (Li et al., 2019).

2. line 312: « Rochefort-en-Yvelines » - Please, delete the extra spaces.

Reviewer 2 Report

The manuscript entitled "Ominoxanthone – the first xanthone linearly fused to a gamma-lactone from Cortinarius ominosus Bidaud Basidiomata. CASE and DFT based structure elucidation" is a very detailed description of the attempt to elucidate the structure of a novel chemical compound found in fungus extract. I like the large number of techniques the authors used to accomplish this task and definitely all modern approaches have been exhaustingly tried, together with molecular modelling approaches. I would say, however, that the task has not been accomplished successfully, even though authors claim that they "elucidated and validated, through CASE and DFT strategies, the structure of ominoxanthone". I would say, that both experimental and computational results point at the structure suggested at the of the study, but with limited certainty. The computational R2 values for the best candidates of 0.999 and 0.996 are very close and likely indistinguishable given the expected accuracy of the DFT methods. Here I would expect some comment about the accuracy of NMR predictions of the DFT methods in general, and in particular this given combination of functional/basis set. Perhaps comments on this problem could be added to the very short Intriduction of this study.

At this moment I'm not convinced that this study provides any new knowledge, even though a lot of work have been put into it. 

Round 2

Reviewer 1 Report

I would like to thank the authors for their detailed comment with references. Despite this I keep ensured that the most adequate approach is the elucidation of the structure using 2 indepent experimental methods at least. Here we have only 2D NMR data (ROESY and HMBC) as a main criterion of correct elucidation. The DFT calculations and literature examples seems beliveable but in my eyes it's not the same as experimental data. If the authors believe that this is the best result that can be achieved in this case, I'm not against the publication.

I have only minor correction:

line 281: Cortinarius sp. - please, use itallic for genus name. In the response it was correct, but in manuscript v.2 it's not the same.

Reviewer 2 Report

Authors of the improved manuscript "Ominoxanthone – the first xanthone linearly fused to a γ-lactone from Cortinarius ominosus Bidaud Basidiomata. CASE and DFT based structure elucidation" answered all of my questions and addressed all of the concerns raised in my first review. I believe that in the current form the manuscript is ready for publication.